# NOISE AND ANATOMY-GUIDED DIFFUSION MODEL FOR REALISTIC CT IMAGE SYNTHESIS

## ABSTRACT

Generative models, specifically Diffusion Models (DMs), have been quite successful in generating high-quality images. However, DMs rely on large-scale training data. In medical imaging, more specifically for computed tomography (CT), these models struggle in accurately reconstructing anatomical structures due to limited training data. This can cause the wrong depiction of organs, which can impact clinical treatment. Some existing models, although guided by anatomical structures, ignore dose-dependent noise, which is critical in real-world scenarios. To tackle this challenge, we propose a novel diffusion model, namely NA-Diff, which is guided by noise from different dose levels and anatomical structures, leveraging a dual conditional diffusion framework. To facilitate large-scale training of DMs on complex structured CT data, we transform natural images emulating realistic CT noises and leverage them for pre-training, followed by fine-tuning on small CT data. Extensive experimental results demonstrate that NA-Diff generates high-fidelity and noise-aware CT images, effectively delineating the organ-of-interest and bridging the gap between synthetic and real CT. [1]

## 1 INTRODUCTION

Computed tomography (CT) serves a crucial role in modern medical imaging, providing detailed anatomical visualization for diagnosis and treatment planning. Each year, more than 80 million CT scans are performed in the United States, and the number keeps growing rapidly Schultz et al. (2020). However, obtaining high-quality CT images is constrained by radiation dose trade-offs, motion artifacts, resolution limitations, and scanner technology. Acquiring large labeled medical imaging datasets can also be challenging due to privacy concerns and expensive annotation processes.

Generative models have shown promise in medical image synthesis, with approaches such as generative adversarial networks (GANs) Goodfellow et al. (2020); Mirza & Osindero (2014) and variational autoencoders (VAEs) Kingma et al. (2019) constructing realistic images. However, these models often suffer from blurring artifacts and difficulties in maintaining fine anatomical structures. In recent years, diffusion models Rombach et al. (2022); Ho et al. (2020) emerged as a strong alternative for image synthesis by modeling a structured denoising process. Unlike GANs, diffusion models (DMs) have more stable training dynamics and produce detailed and diverse outputs. DMs have been successfully adapted to various domains, including medical imaging Bhattacharya et al. (2024); Konz et al. (2024); al Nomaan Nafi et al. (2024); Munia & Imran (2025), specifically, conditional ones over their unconditional counterparts. Conditional DMs rely on a large amount of labeled data for effective training. To mitigate this dependency, alternatives such as self-supervised learning, weak supervision, data augmentation, and synthetic data generation are being explored Liu et al. (2024); Oh & Jeong (2024). In these scenarios, generated outputs may appear anatomically plausible but lack clinical precision, while augmentation can introduce hallucinations or structural distortions, making them inappropriate for medical applications.

For image synthesis tasks, generative models Goodfellow et al. (2020); Mirza & Osindero (2014); Rombach et al. (2022); Ho et al. (2020) portray remarkable performances, leveraging a satisfactory amount of large data for image generation. Yet, such extensive datasets are not accessible to the medical imaging domain or in real-world clinical environments, which limits the scalability of different approaches Song et al. (2023); Chen et al. (2024); al Nomaan Nafi et al. (2024). In response to

---

[1]The code will be made available upon acceptance.

the limited data issue, pre-training generative models on natural image datasets (e.g., ImageNet) can be a possible solution. Regardless, it leads to domain shift, as the characteristics (e.g., Hounsfield units, noise patterns) of medical images (e.g., CT images) differ significantly from those of natural images. This domain shift can degrade the quality of synthesized CT images and limit their clinical utility Zhang et al. (2018b). Another line of work introduces annotation-guided CT generation Venkatesh et al. (2024); Bhattacharya et al. (2024); Yang et al. (2024); Konz et al. (2024) to infuse shape constraints and organ-specific priors into the synthesis process. Textual prompts, combined with anatomical masks, have been utilized to generate high-quality 3D CT scans Xu et al. (2024). However, the model is built specifically for 3D volumetric CT synthesis and does not support 2D image generation. Furthermore, medical image synthesis with DMs can produce unrealistic artifacts or hallucinations Cho et al. (2025), which can potentially pose severe risks in subsequent image-based decision-making. Hallucination may arise when the model depends only on anatomy-guided conditioning without noise awareness, while focusing solely on noise guidance may lead to the loss of anatomical fidelity. To address the aforementioned challenges, we propose a novel dual conditional diffusion-based realistic CT image synthesis approach with CT noise and anatomy guidance, leveraging a large number of diverse natural images with emulated CT noise at various dose levels. The model is pre-trained using emulated CT noise guidance, and this is followed by fine-tuning with a small number of real CT images guided by dose-aware noise and anatomy maps. First, we generate CT-like noise–emulated natural images using our proposed strategy, so we can pre-train on abundant natural images that mimic CT noise characteristics. Then, during fine-tuning on real CT data, we introduce a dual-conditioning setup with both a noise map (dose-aware) branch and an anatomy-guidance branch for CT image synthesis. This combination of synthetic noise-emulated pretraining and dual noise–anatomy conditioning, to our knowledge, has not been explored in prior CT diffusion work and goes beyond a simple diffusion architecture modification.

The contributions of the paper are summarized as follows:

- A novel framework for CT image synthesis guided by noise and anatomy maps using a dual conditioning technique for preserving realistic noise-aware anatomical properties.

- Emulation of CT images and CT noise-emulated natural images at different dose levels with corresponding noise maps. This procedure is effective and adapts to varying radiation exposures.

- A pre-training strategy utilizing CT noise-emulated natural images at different CT dose levels to learn useful low-level and high-level features from the large dataset, enabling better performance with limited CT image data.

- Extensive evaluations, including external validation, across a range of tasks and metrics, demonstrating consistent and superior performance of NA-Diff in generating realistic CT.

## 2 RELATED WORK

### 2.1 DIFFUSION MODELS

Diffusion models have revolutionized generative modeling, demonstrating unprecedented capabilities in image generation by progressively denoising random noise to synthesize realistic visuals Ho et al. (2020); Song et al. (2020); Song & Ermon (2019). Recent conditional variants refined this approach, enhancing synthesis control by integrating conditions like textual prompts, segmentation masks, or style embeddings Dhariwal & Nichol (2021); Ho & Salimans (2022); Lugmayr et al. (2022); Saharia et al. (2022). Despite significant progress, unconditional models offer limited control over content and structure, while conditional models are often dependent on the availability and quality of external conditioning data. Recent works have addressed these challenges by introducing diffusion Ho et al. (2020); Peebles & Xie (2023) with image conditioning Zhu et al. (2025); Konz et al. (2024); Zhang et al. (2023); Shi et al. (2024), achieving more robust semantic guidance and motivating further exploration in domains that demand precise structural fidelity, such as medical image synthesis.

### 2.2 DIFFUSION FOR MEDICAL IMAGE SYNTHESIS

Latent diffusion-based model MediSyn Cho et al. (2025) effectively generates diverse medical images across multiple imaging modalities using textual guidance. However, text-based conditioning

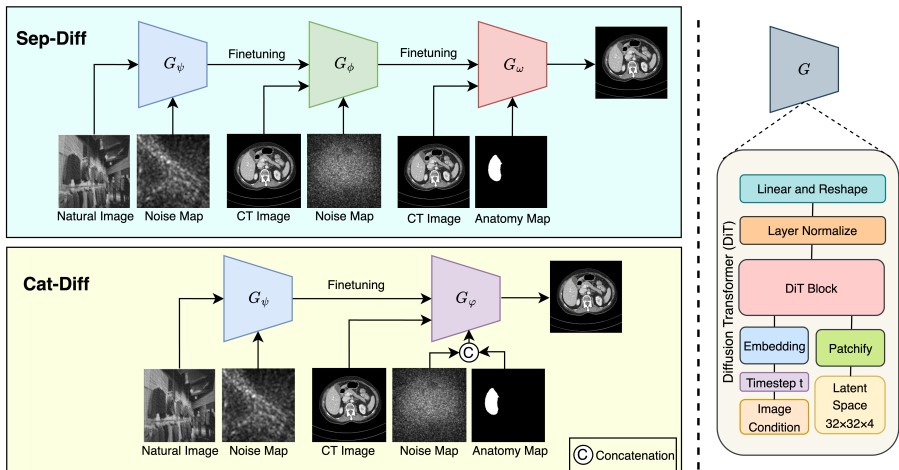

Figure 1: Overview of the proposed NA-Diff framework, illustrating its two variants: Sep-Diff and Cat-Diff. In Sep-Diff variant, the model is trained in three stages leveraging separate conditions while in Cat-Diff, the model utilize dual conditions with two stage training. $G_\psi$, $G_\phi$, $G_\omega$ and $G_\varphi$ denotes pre-training model on CT noise-emulated natural images with noise map guidance, fine-tuning model on CT images with noise map guidance, fine-tuning model on CT images with anatomy guidance, and fine-tuning CT images with corresponding noise map and anatomy guidance both, respectively.

is inherently imprecise for fine-grained anatomical control, often resulting in unrealistic anatomical structures Zhang et al. (2024); Chaichuk et al. (2025). While ControlNet Zhang et al. (2023), initially developed for natural image generation, has a mechanism for adding spatial conditioning through structural inputs, making it adaptable for medical domains where spatial fidelity is critical. Yet, direct application to medical domains may suffer from domain shift, limiting anatomical realism and clinical utility without retraining on medical-specific data. Recent anatomy-aware diffusion frameworks embed structural guidance (e.g., segmentation masks, anatomical labels, semantic layouts) directly into the denoising trajectory to ensure radiologically faithful synthesis. Seg-Diff Konz et al. (2024) proposed a segmentation-guided diffusion model that generates medical images by conditioning on segmentation masks at every denoising step. Using cross-attention or dual-stream architectures, some techniques generate image–mask pairs or condition on latent anatomy embeddings, achieving spatial coherence across both 2-D slices and volumetric reconstructions Xing et al. (2024); Zhang et al. (2025); Zheng et al. (2024); Bhat et al. (2025); Jiang et al. (2025); Mao et al. (2025); Bose et al. (2025). Their shared strategy shows that anatomy-guided conditioning markedly boosts structural realism and downstream segmentation performance in CT and related modalities. However, they only depend on precise anatomy inputs at inference and ignore noise-aware conditioning, leaving the problem of converting noisy CT projections into high-quality volumes largely unsolved.

## 2.3 Context-Aware Multi-Guidance Diffusion

Relying on a single conditioning process in diffusion models is sometimes flawed, as generating target-like images may require using more than one type of guidance such as structural, and style guidance Konz et al. (2024); Xing et al. (2024); Mao et al. (2025). Incorporating multiple guidance can be positively effective, as it enables the model to capture both stylistic attributes and semantic content with greater fidelity Krishna et al. (2024); Zhu et al. (2023); Shen et al. (2024). For CT image synthesis, both noise conditioning (e.g., dose-aware noise maps) and anatomical guidance are important to enable realistic simulation. The model DCDiff Shen et al. (2024) generates clean CT images with reduced metal artifacts. However, the model lacks awareness of different dose levels and may struggle to generate images with varied noise characteristics corresponding to each dose. To the best of our knowledge, no existing model is able to synthesize CT images that are simultaneously dose-aware and structure-aware, capturing both noise characteristics and anatomical fidelity within a unified framework. The conditioning strategy also varies across different models. For encoding images, most existing models use a VAE-based encoder Kingma et al. (2019) for processing input or

conditioning information which is efficient and flexible guidance in the latent space Ho et al. (2020); Zhang et al. (2023); Esser et al. (2021); Nie et al. (2024); Peebles & Xie (2023). In contrast, for some cases, CLIP (Contrastive Language-Image Pretraining) Radford et al. (2021) can be used for image embeddings instead of VAE, as CLIP's ability to extract semantically rich, global features from images or text allows more flexible and effective conditioning for diffusion models Zhu et al. (2025).

## 3 METHOD

Fig. 1 illustrates our proposed diffusion-based CT image generation framework NA-Diff. NA-Diff includes two variants: (i) separate conditioning with noise map and anatomy mask (Sep-Diff), and (ii) concatenate conditioning with noise map and anatomy mask (Cat-Diff). Sep-Diff consists of a three-stage training process that systematically enables the synthesis of new CT images. The objective is to synthesize CT images conditioned on noise maps or anatomy maps. However, we cannot simultaneously condition the model on both the noise map and the corresponding anatomy mask, as each stage of the model supports only a single conditioning input. In light of this limitation, we introduce Cat-Diff, a two-stage training process that leverages both noise maps and corresponding anatomy masks to synthesize CT images.

### 3.1 DIT

DiT Peebles & Xie (2023) is a transformer-based diffusion model that operates on latent patches for image generation task, achieving state-of-the-art image quality on benchmarks. It uses a latent diffusion framework and processes image representations as patches following Vision Transformers (ViTs) Dosovitskiy et al. (2020). Given an input image $x_0$ and a corresponding condition $y$ (e.g., text-guided, noise map, or anatomy-guided

Table 1: Details of the NA-Diff model variants.

| Type | Model | Description |
|------|-------|-------------|
| **Sep-Diff** | NA-Diff-A | Training $G_\phi$ with CT images conditioned on noise map data. |
| | NA-Diff-B | Training $G_\psi$ with natural image data, followed by fine-tuning ($G_\phi$) with CT images conditioned on noise map data. |
| | NA-Diff-C | Fine-tuning ($G_\omega$) using NA-Diff-A, with CT images conditioned on anatomy map data. |
| | NA-Diff-D | Fine-tuning ($G_\omega$) using NA-Diff-B, with CT images conditioned on anatomy map data. |
| **Cat-Diff** | NA-Diff-E | Training $G_\varphi$ with CT images conditioned on noise map and corresponding anatomy map. |
| | NA-Diff-F | Training $G_\psi$ with natural image data, followed by fine-tuning ($G_\varphi$) with CT images conditioned on noise map and corresponding anatomy map. |

image), the model learns to reconstruct $x_0$ through iterative denoising process: $p_\theta(\hat{x}_0|x_0, y)$ where $\theta$ represents the model parameters. Like other diffusion models Rombach et al. (2022); Ho et al. (2020), DiT follows a forward process that gradually adds Gaussian noise to the input image and a reverse process that aims to denoise and reconstruct the original image conditioned on a noise map or anatomy map. The model uses self-attention mechanisms and multi-head attention layers to model long-range dependencies within the image. This allows the network to process complex textures and structures efficiently. During DiT training, a noise prediction loss is minimized: $L_{(\theta)} = \mathbb{E}_{x_0, y, t, \epsilon} \left[ \|\epsilon_\theta(x_t, y) - \epsilon\|_2^2 \right]$, where $\epsilon_\theta(x_t, y)$ is the predicted noise from the diffusion transformer, $\epsilon \sim \mathcal{N}(0, \mathbf{I})$ is the true noise sampled from a Gaussian distribution, and $\mathbb{E}$ represents taking the mean over these random variables, ensuring generalization across varying noise conditions.

### 3.2 NATURAL IMAGE PRETRAINING

**Emulation of CT noise in natural images:** Natural images are by default of high quality. We can consider them equivalent to full-dose CT images. An RGB image is first converted to grayscale and we map it to CT water density. For simplicity, we assume $1 \times 1$ pixel size and projection angles 0-180°. The raw projection data (sinogram) $r$ is generated by performing parallel beam projections (Radon transform). The generated sinogram is fed to inverse Radon (iRadon) transform to reconstruct the original grayscale CT-like full-dose image $x_{fd}^{nat}$. To reduce spectral leakage and enhance interpretation, we select 'Hann' filter and linear interpolation for iRadon. To match with our CT dataset, we generate lower-dose image ($x_{ld}$) at the same dose level ($ld$). Assuming the number

of photons $I_0 = 1e^5$ emitted at full dose and water attenuation of $\mu = 0.02$, we can then calculate the number of mean transmitted photons for the raw data as $I_{raw} = ld \cdot I_0 \cdot \exp(-\mu r)$. Poisson quantum noise is inserted into the raw project data at the target dose level. Therefore, the noisy projection we obtain as:

$$I_{noisy} = -log(Poisson(I_{raw})/(ld \cdot I_0))/\mu. \tag{1}$$

Using iRadon, the noisy image $x_{ld}^{nat}$ is then reconstructed from the noisy projection data at the target dose level. Following the approach described in Sec. 3.3 for CT images, natural images are also simulated at additional dose levels, and corresponding noise maps are constructed (Fig. 2).

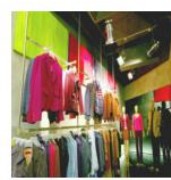 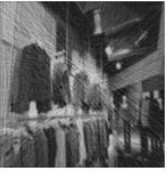 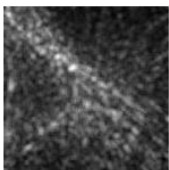

RGB image      Noisy image      Noise map

Figure 2: Sample noisy image and corresponding noise map constructed from a natural image by emulating CT noise.

In the pretraining phase, input natural images are denoted $x_0^{nat}$, and corresponding noise maps are denoted by $y_{noise}^{nat}$. To synthesize a refined natural image $\hat{x}_0^{nat}$, we use the DiT model conditioned on $y_{noise}^{nat}$, replacing the text conditioning with an image conditioning strategy. For the pre-training stage of both Sep-Diff and Cat-Diff variants, $x_0$ and $y$ are mapped into the latent space using a pre-trained VAE Kingma et al. (2019). Those are patchified and linearly embedded to create token representations: $z_{x_0} = W_x x_0 + E_{pos}$ and $z_y = W_y y + E_{pos}$; where $W_x, W_y$ are learned embedding matrices, $E_{pos}$ is the positional encoding, and $z_{x_0}, z_y$ are the patch embeddings. The timestep embedding is combined with the conditional noise embedding $c = f_\theta(t) + z_y$; $f_\theta(t)$ is an MLP-based mapping timestep $t$ to an embedding, and $c$ is the conditioning vector that modulates the model.

Each Transformer block uses AdaLN-Zero, where the modulation parameters $(\gamma_t, \beta_t, \alpha_t)$ are predicted from the conditioning vector $c$:

$$(\gamma_t, \beta_t, \alpha_t) = \text{MLP}(\text{LayerNorm}(c)), \tag{2}$$

where $\gamma_t$ and $\beta_t$ are feature-wise scaling and shifting parameters, and $\alpha_t$ is a learned residual gating parameter that controls the strength of each residual branch. Given these parameters, a DiT block updates the hidden representation $x$ as:

$$x = x + \alpha_t \cdot \text{MSA}(\gamma_t \cdot \text{LayerNorm}(x) + \beta_t), \tag{3}$$

$$x = x + \alpha_t \cdot \text{MLP}(\gamma_t \cdot \text{LayerNorm}(x) + \beta_t), \tag{4}$$

where the initial hidden state is the image token sequence $x = z_{x_0}$; that is, $z_{x_0}$ is the tokenized representation of the input image, and $x$ is refined iteratively by the stacked Transformer blocks. The model predicts the noise $\hat{\vartheta}(x_t, t, y)$ and the output AdaLN-Zero parameters are computed as:

$$(\gamma'_t, \beta'_t) = \text{MLP}(\text{LayerNorm}(c)), \tag{4}$$

and the noise prediction is generated by:

$$\hat{\vartheta} = \text{MLP}(\gamma'_t \cdot \text{LayerNorm}(x) + \beta'_t). \tag{5}$$

The denoised image $\hat{x}_0$ is computed during sampling using the standard DM reconstruction:

$$\hat{x}_0 = \frac{x_t - \sqrt{1 - \bar{\alpha}_t}\, \hat{\vartheta}}{\sqrt{\bar{\alpha}_t}}, \tag{6}$$

where $\bar{\alpha}_t$ is the cumulative product of the variance schedule in the forward diffusion process and $\hat{x}_0$ is the final denoised image. Also, $\gamma'_t$ and $\beta'_t$ are dynamically learned based on $y$ and $t$.

### 3.3 NOISE-GUIDED CT FINE-TUNING

**Low dose simulation:** Following Imran et al. (2021), we use the quantum noise properties of low-dose images, assuming a linear relationship through reconstructions with the relative dose levels.

Given real full-dose and lower-dose scans, CT scans can be simulated at any arbitrary dose level ($d$). With the full-dose noise variance $\sigma_{fd}^2$, an image $x_d$ can be formed by adding zero-mean independent noise to the full-dose image $x_{fd}$.

$$x_d = x_{fd} + x_{noise},$$
$$\text{where, } x_{noise} \sim \mathcal{N}(0, (1/d - 1)\sigma_{fd}^2). \tag{7}$$

In order to estimate $\sigma_{fd}^2$, we use the difference between $x_{fd}$ and the available lower-dose image $x_{ld}$. We are then able to synthesize images at any arbitrary dose level by determining $a$ from $1 + (1/d - 1)a^2 = 1/d$.

**Noise estimation from single slice/image:** After obtaining noise realization from a single slice of the given $x_{fd}$ and $x_{ld}$, it can be scaled to derive the noise of the desired dose level, i.e., $n = b \cdot (x_{ld} - x_{fd})$. And we can find $b$ such that

$$b^2 \cdot (\frac{1}{d} - 1) = \frac{1}{d}. \tag{8}$$

Ideally, noise realizations should be uncorrelated with noise at the routine dose, but correlated with noise at all other dose levels. It can be verified that noise standard deviation (std) in uniform regions of an image matches the expected noise levels in HU. It is, therefore, viable to use noise std map as a means of assessing the quality of CT images at arbitrary dose levels. Using a sliding window with window size $k \times k$ over the noise realization $n$ in the image space at $m \times m$ resolution, the std map is calculated as:

$$y_n^{ct} = \sqrt{\frac{1}{k^2} \sum^{m \times m} (n(k,k))^2}. \tag{9}$$

With the increase in the window size, the noise std scale (HU) is shrunk. We choose $k = 5$ in order to keep maximum scale gaps in different dose levels.

In Sep-Diff variant, with the simulated CT images and the corresponding noise (std) maps, the natural image pre-trained model is fine-tuned on $(G_\phi)$. Similar to pre-training of $G_\psi$, $G_\phi$ is fed by an input CT image $x_0^{ct}$, and its corresponding noise map $y_{map}^{ct}$ for conditioning. The training process is similar to the pre-training phase, and the final layer reconstructs the input CT image $\hat{x}_0^{ct}$. However, unlike the DiT approach, which is used in the pre-training phase, the conditioning strategy for CT images corresponding to the map leverages a pre-trained CLIP encoder instead of VAE. Instead of directly using pixel-level features as a condition, $y_{map}^{ct}$ is first processed using the CLIP processor and mapped into a global semantic embedding using the CLIP encoder. This embedding captures high-level semantic information, which is then used to guide the generative process. Using the diverse dose level images in training, Sep-Diff is enabled to learn the realistic variation in the image quality, retaining textures. For a single diagnostic task, higher-dose, lower-noise CT images typically improve performance. However, radiation dose cannot be increased freely in practice, especially in screening, pediatrics, and repeated follow-ups. Multi-center datasets also contain diverse protocols and dose levels, leading to substantial variation in noise and texture. Our aim in modeling dose-aware, higher-dose, lower-noise CT images is not to promote increased radiation, but to provide a controllable way to simulate different dose (noise) levels from the same anatomy. This enables systematic study of how downstream tasks behave across dose levels, generation of virtual high-dose images from low-dose scans without extra patient dose, and harmonization of heterogeneous datasets by mapping images to a standardized noise level.

### 3.4 ANATOMY-GUIDED CT FINE-TUNING

In Sep-Diff, for the second stage of fine-tuning, we introduce anatomy masks as additional guidance. The main reason for incorporating anatomical guidance in CT image synthesis is to enforce anatomical constraints in the generated images, and this helps the model to capture the fundamental structures and distribution of CT images. The anatomy mask is derived from the simulated low-dose CT images using a segmentation model Ward & Imran (2025). The training process is similar to the first stage of fine-tuning in Sep-Diff (i.e., $x_0^{ct}$ as CT image and $y_{mask}^{ct}$ as anatomy guidance). In this phase, we use the DiT layout, modifying the condition strategy, and the final layer reconstructs the denoised image. It maintains anatomical plausibility, followed by the anatomy mask.

Table 2: Consistency evaluation of image quality and liver segmentation between synthetic and input CT images. $IQA_{real}$ and $IQA_{gen}$ indicate the average IQA scores obtained for the input CT test set and the NA-Diff variant generated CT images, respectively. $m$ is the original segmentation liver mask, and $m_{gen}^{pred}$ are the predicted masks of the segmentation model from the generated images. The best and second-best results are **bolded** and underlined respectively.

| NA-Diff | Condition | | Image quality | | | Segmentation | | | | |
|---|---|---|---|---|---|---|---|---|---|---|
| | Noise | Anatomy | $IQA_{real}$ | $IQA_{gen}$ | $\Delta_{IQA}\downarrow$ | $Dice(m_{gen}^{pred},m)\uparrow$ | $IoU(m_{gen}^{pred},m)\uparrow$ | $HD(m_{gen}^{pred},m)\downarrow$ | $Recall(m_{gen}^{pred},m)\uparrow$ | $Precision(m_{gen}^{pred},m)\uparrow$ |
| NA-Diff-A | ✓ | ✗ | 1.940 | 2.260 | 0.320 | 0.692 | 0.532 | 50.730 | 0.644 | 0.781 |
| NA-Diff-B | ✓ | ✗ | 1.940 | 2.190 | 0.250 | 0.711 | 0.5548 | 46.725 | 0.687 | 0.768 |
| NA-Diff-C | ✗ | ✓ | 1.940 | 2.150 | 0.210 | 0.774 | 0.633 | 36.179 | 0.776 | 0.791 |
| NA-Diff-D | ✗ | ✓ | 1.940 | 2.100 | 0.160 | **0.779** | **0.640** | **30.494** | **0.797** | 0.782 |
| NA-Diff-E | ✓ | ✓ | 1.940 | 2.110 | 0.170 | 0.725 | 0.572 | 49.135 | 0.669 | **0.813** |
| NA-Diff-F | ✓ | ✓ | 1.940 | 2.080 | **0.140** | 0.775 | 0.633 | 31.997 | 0.796 | 0.781 |

## 3.5 DUAL CONDITIONING DIFFUSION

In the Cat-Diff variant, we introduce dual conditioning for the diffusion process that provides one stage of fine-tuning. In contrast, the Sep-Diff variant follows a two-stage fine-tuning strategy: initially, the model is fine-tuned using noise map-guided CT images, and subsequently, it is further fine-tuned with anatomy maps as conditions, utilizing the same CT images from the first stage of fine-tuning. When fine-tuning the model in two separate stages, first with noise maps and then with anatomy maps, a limitation becomes evident. During the initial stage of fine-tuning, the model learns to rely only on the noise map, since it has not seen anatomy-based conditioning. However, after the second fine-tuning using anatomy maps, the model's dependency shifts. Its ability to respond to noise variation decreases, and any influence from the noise map becomes largely random and un-controlled. To overcome this challenge and ensure the model can utilize both noise and anatomy maps, we propose the Cat-Diff variant, where the dual conditioning approach allows the model to incorporate both forms of guidance during the diffusion process. As a result, the final model can benefit from the complementary information provided by both noise and segmentation maps, enabling more flexible and controlled image synthesis. As shown in Fig. 3, in the Cat-Diff approach, we utilize two conditions by encoding both with a pre-trained CLIP encoder, rather than relying on VAE-based conditioning. Specifically, each conditioning input $y$ (e.g., a noise-level map $y_{\text{map}}$ and a segmentation mask $y_{\text{mask}}$) is encoded into a fixed-length global embedding vector of dimension 768 through CLIP. By conditioning on CLIP-derived global embeddings, our model leverages high-level, semantically meaningful representations and enables effective guidance from both noise-level and structural modalities.

We ensure the generated CT data is reliable by conditioning the model on both noise maps and anatomy masks. The noise map guides the model to reproduce realistic noise patterns, while the anatomy mask preserves the underlying structures. As the model must satisfy both constraints, the generated images maintain correct anatomy and visually plausible CT noise characteristics. Different noise levels are clinically relevant, although higher-dose, lower-noise CT scans produce clearer images, radiation dose cannot be increased freely due to safety concerns, and real multi-center datasets often contain scans with very different noise levels. Being able to simulate multiple noise conditions from the same anatomy is therefore useful for understanding how downstream tasks behave across arbitrary dose levels and for harmonizing heterogeneous datasets.

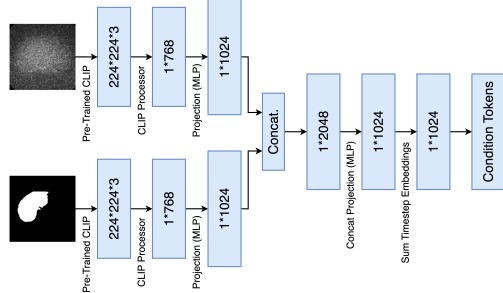

Figure 3: Dimensionality flow of dual conditioning. Condition maps and masks are encoded by a pre-trained CLIP model into global features, projected and fused with timestep embeddings, then combined with image tokens for the transformer-based diffusion model.

## 4 EXPERIMENTS & RESULTS

### 4.1 IMPLEMENTATION DETAILS

**Natural Image Data:** We utilize the indoor scene Indoor Scene Recognition dataset Quattoni & Torralba (2009) and generate CT-like images emulating CT noises at quarter dose. From 15,620

images, we selected 4,000 images with complex-structured and intricate details. Following Sec. 3.2, a total of 20,000 natural images and their corresponding noise maps are generated at different dose levels (i.e., 5%, 10%, 25%, 50%, and 75%).

**SDCT Data:** SDCT refers to the six-dose computed tomography, representing our emulated CT dataset. For CT images, we use 40 abdomen scans (25% and 100% doses) from the Low Dose CT Image and Projection Data McCollough et al. (2020) — 20 for training and 20 for testing. After simulation of additional dose levels, a total of 13,314 (e.g., 6696 for training and 6618 for testing) CT images and noise map pairs are obtained. All CT images in our figures are displayed using a standard soft-tissue window, with a window width (WW) of 400 HU and a window level (WL) of 40 HU. The anatomy masks are obtained from the simulated low-dose CT images using the liver segmentation model Ward & Imran (2025). All 13,314 simulated low-dose CT images are processed to generate the corresponding liver segmentation masks referred to as anatomy maps. Specifically, liver segmentation models are well-established and generally produce reliable masks. Inaccurate or noisy masks can directly degrade anatomy-guided synthesis and make it difficult to evaluate the method itself. Therefore, we focus on the liver organ for this study. However, the approach is not liver-specific as it can be extended to other organs (e.g., pancreas) with the availability of high-quality organ masks with relative dose levels.

**Training:** We trained the DiT model from scratch, as our task differs from ImageNet-trained DiT models. For training NA-Diff, we start with the DiT-L/4 model ($256 \times 256$ resolution). With a patch size of 4, L/4 processes a total of 1024 tokens after patchifying the $32 \times 32 \times 4$ input latent. To process condition images, we use a pre-trained CLIP encoder to obtain global feature embeddings. Natural images are pre-trained for 25 epochs. For CT image synthesis, we train the models for 100 epochs with noise conditioning, 100 epochs with

Table 3: Quantitative comparison of generative performance using CFID, KID, CMMD, and LPIPS across state-of-the-art baseline and our anatomy-guided NA-Diff variants. Lower scores indicate better fidelity, perceptual quality, and conditional consistency. ∗ denotes that the model is trained with our CT image data conditioned on the anatomy mask.

| Methods | Performance metrics | | | |
|---|---|---|---|---|
| | CFID ↓ | KID ↓ | CMMD ↓ | LPIPS ↓ |
| Seg-Diff Konz et al. (2024) | 249.769 | $122.358 \pm 13.406$ | 0.973 | $0.703 \pm 0.281$ |
| Seg-Diff∗ Konz et al. (2024) | 111.938 | $103.993 \pm 11.255$ | 0.670 | $0.664 \pm 0.145$ |
| ControlNet∗ Zhang et al. (2023) | 124.418 | $104.313 \pm 8.5501$ | 0.620 | $0.580 \pm 0.117$ |
| NA-Diff-C | 89.295 | $57.601 \pm 5.823$ | 0.541 | $0.387 \pm 0.102$ |
| NA-Diff-D | **53.295** | $\mathbf{19.039 \pm 2.853}$ | **0.333** | $\mathbf{0.382 \pm 0.103}$ |

anatomy conditioning, and 150 epochs with dual conditioning. The models are trained on an Intel (R) Xeon (R) w7-2475X, 2600MHz machine with dual NVIDIA A4000 GPUs (16X2=32GB).

**Baseline and Compared Methods:** Considering no existing generative models using CT noise-emulated natural images with noise map conditioning and anatomy guidance, we compare different variant types of the NA-Diff model (Table 1). Direct comparison with our dual condition-guided approach is not entirely applicable; however, we include Seg-Diff Konz et al. (2024) (e,g, the state-of-the-art anatomy-guided diffusion model for CT images) and ControlNet Zhang et al. (2023), as the representative anatomy-guided diffusion baseline and trained with our SDCT dataset, only leveraging anatomy guidance as conditions. For a fairer assessment, we report results from our only anatomy-conditioned variant alongside the dual-conditioned model. In addition, we have shown comparisons with the baseline using the LiTS Bilic et al. (2023) dataset.

**Noise and Anatomy Preservation Evaluation:** NA-Diff model variants are evaluated by predicting diagnostic image quality assessment (IQA). We use a pre-trained IQA model Rifa et al. (2025b) to provide a measure of noise level estimation for the generated images. We also utilize a pretrained segmentation model Ward & Imran (2024) for generating liver masks from the generated images.

Table 4: Consistent IQA scores are observed for NA-Diff-D and NA-Diff-F compared to training data across different dose levels.

| Source | Dose Level (%) | | | | | |
|---|---|---|---|---|---|---|
| | 5 | 10 | 25 | 50 | 75 | 100 |
| Train set | 1.500 | 1.450 | 1.870 | 2.110 | 2.290 | 2.410 |
| NA-Diff-D | 2.090 | 2.020 | 2.040 | 2.140 | 2.140 | 2.140 |
| NA-Diff-F | 1.910 | 1.960 | 2.120 | 2.140 | 2.160 | 2.190 |

**Image Quality:** We use the popular metric CFID Parmar et al. (2022) and KID Bińkowski et al. (2018) for image realism and CMMD Jayasumana et al. (2024) to provide an unbiased and semantically meaningful evaluation of image quality using CLIP features, and LPIPS Zhang et al. (2018a) for perceptual similarity.

Table 5: Ablation study demonstrating the importance of pre-training and dual conditioning. The best and second-best results are **bolded** and underlined respectively. Nat. Pre-Train denotes training $G_\psi$ with natural image data.

| Methods | Nat. Pre-Train | Noise | Anatomy | CFID ↓ | KID ↓ | CMMD ↓ | LPIPS ↓ |
|---------|:---:|:---:|:---:|:---:|:---:|:---:|:---:|
| | | | | | **Performance metrics** | | |
| NA-Diff-A | ✗ | ✓ | ✗ | 245.342 | $209.733 \pm 10.736$ | 1.748 | $0.415 \pm 0.097$ |
| NA-Diff-B | ✓ | ✓ | ✗ | 152.768 | $102.657 \pm 7.917$ | 0.754 | $0.383 \pm 0.098$ |
| NA-Diff-C | ✗ | ✗ | ✓ | 89.298 | $57.601 \pm 5.823$ | 0.541 | $0.387 \pm 0.102$ |
| NA-Diff-D | ✓ | ✗ | ✓ | **53.295** | $\mathbf{19.039 \pm 2.853}$ | 0.333 | $0.382 \pm 0.103$ |
| NA-Diff-E | ✗ | ✓ | ✓ | 78.373 | $40.328 \pm 4.512$ | 0.514 | $0.389 \pm 0.094$ |
| NA-Diff-F | ✓ | ✓ | ✓ | 64.638 | $24.811 \pm 3.228$ | **0.321** | $\mathbf{0.376 \pm 0.099}$ |

Figure 4: Qualitative comparison among generated images from different NA-Diff variants.

## 4.2 RESULTS AND DISCUSSION

As reported in Table 2, we perform comparisons among the variant types of NA-Diff generated images, leveraging the noise map, the anatomy mask, or both, conditioned. The generated images and the simulated CT test images are assessed by IQA scores (0-4, higher is better) Lee et al. (2025). The pretrained IQA model Rifa et al. (2025b) was trained on a different window-level CT dataset, yet it can approximately estimate the noise level in our generated images. NA-Diff-F has the smallest IQA gap ($\Delta_{IQA}$) among all, demonstrating the effectiveness of the proposed noise-guided training processes. Additionally, the larger $\Delta_{IQA}$ for the baseline Model A (without natural image pre-training) highlights the importance of CT noise-emulated pre-training in NA-Diff. Moreover, consistent with the dose-IQA relationship, NA-Diff-F generated images are of better quality with the increase in relative dose levels (Table 4), indicating the intrinsic noise aware dose level understanding over NA-Diff-D. In terms of segmentation in Table 2, we show a comparison between the original mask $m$ and masks $m_{gen}^{pred}$ predicted for the generated images. The NA-Diff-D and NA-Diff-F model achieves comparatively high scores across all metrics, which indicates good anatomical consistency.

In Table 3, we compare our anatomy-guided model variants (i.e, NA-Diff-C and NA-Diff-D) with the baseline model Seg-Diff Konz et al. (2024). Compared to the anatomy-conditioned baseline Seg-Diff*, NA-Diff-D achieves substantial improvements across all metrics, where CFID improves by 52.4%, KID by 81.7%, CMMD by 50.4%, and LPIPS by 42.5%. When benchmarked against the ControlNet* baseline, NA-Diff-D achieves even larger relative gains, with CFID reduced by 57.2%, KID by 81.8%, CMMD by 46.3%, and LPIPS by 34.1%. These results show that our approach can generate anatomy-guided CT images with better fidelity, perceptual quality, and consistency compared to previous methods. The results of the ablation study are shown in Table 5. Across all metrics,

Table 6: Segmentation performance on the downstream task using real data and a combination of real plus synthetic data. The models are trained either on real data only or on a mixture of real and synthetic CT images (paired with masks), and evaluated on the first 1000 cases from the test set (using real images and masks only). Incorporating synthetic data leads to comparable improvements in Dice and IoU across both U-Net and TransUNet architectures.

| Methods | Description | Dice | IoU |
|---------|-------------|------|-----|
| U-Net Ronneberger et al. (2015) | Real Data | 0.874 | 0.777 |
| | Real + Synthetic Data | 0.876 | 0.779 |
| TransUNet Chen et al. (2021) | Real Data | 0.872 | 0.773 |
| | Real + Synthetic Data | 0.877 | 0.781 |

natural image pre-training relatively improves performance compared to no pre-training. Comparing

NA-Diff-D (with pre-training) to NA-Diff-C (without pre-training), we observe substantial improvements in CFID by 40.3%, KID by 66.9%, CMMD by 38.4%, and LPIPS by 1.3%.

For dual conditioning, NA-Diff-F's CFID and KID are higher than NA-Diff-D because CFID is conditioned only on anatomy for calculation, favoring NA-Diff-D, and KID reflects greater sample diversity. However, NA-Diff-F achieves 3.8% lower CMMD and 1.5% lower LPIPS, indicating better conditional consistency and perceptual similarity with dual conditioning. As shown in Table 6, adding synthetic data provides comparable gains in Dice and IoU, which demonstrate the potential of our synthetic CT images for downstream segmentation tasks.

For completeness, we report the Dice scores with 95% confidence intervals: U-Net (Real: 0.874 [0.857–0.892], Real+Synthetic Data: 0.876 [0.858–0.893]) and TransUNet (Real: 0.872 [0.855–0.890], Real+Synthetic Data: 0.877 [0.860–0.894]), showing that the improvements are consistent and synthetic data does not degrade segmentation performance.

Furthermore, the qualitative results are presented in Fig. 4, where our NA-Diff-F model generates the CT images according to the noise-aware semantic mask. In addition, as evidenced from Table 7, our NA-Diff-D model outperforms the baseline when evaluated on the LiTS dataset. We compare the original segmentation masks with those predicted from the generated images (i.e., segmentation masks constructed by applying a segmentation model Ward & Imran (2025) to the generated images). This evaluation demonstrates that our method effectively preserves anatomical structures in the synthesized CT images in comparison to the baseline model. Overall, NA-Diff-F performs well while following both noise and anatomy guidance. However, if we focus only on anatomy guidance, NA-Diff-D gives favorable results. In addition to the improvement in image generation, NA-Diff can be more resource-efficient compared to the baseline Seg-Diff. Seg-Diff relies on a high-resource setup with $4 \times$ A6000 GPUs (48 GB each), whereas our NA-Diff models were trained using only $2 \times$ A4000 GPUs (32 GB total). Despite using far fewer resource requirements, NA-Diff-D still achieves higher Dice, IoU, HD, recall, and precision. Regarding data efficiency, Seg-Diff is trained on approximately $10 \times$ more real CT images than our NA-Diff model (11000 vs 1116). Considering the scarcity of high-quality medical datasets, our NA-Diff model reduces reliance on real CT data by leveraging large natural-image collections through our CT-like noise-emulation and pretraining strategies, enabling strong synthesis performance with far less real CT data and lower computational cost.

Table 7: The models are fine-tuned (with 3000 random sample images) and evaluated (with 500 random sample images) with segmentation guidance using LiTS Bilic et al. (2023) dataset. Our synthetic datasets show superior performance to the baseline. ∗ denotes that the model is trained with our CT image data conditioned on the anatomy mask.

| Methods | Performance metrics | | | | |
|---------|------|------|--------|--------|-----------|
|         | Dice | IoU  | HD     | Recall | Precision |
| Seg-Diff* | 0.911 | 0.838 | 22.629 | 0.864 | 0.968 |
| NA-Diff-C | 0.910 | 0.836 | 21.382 | 0.866 | 0.970 |
| NA-Diff-D | **0.920** | **0.853** | **20.508** | **0.872** | **0.974** |

## 5 CONCLUSIONS

In this study, we present a novel noise and anatomy-guided diffusion model, NA-Diff, that can generate realistic CT images with dose-level aware noise and liver anatomy map guidance. Our innovative CT noise-emulated natural image pre-training helps capture complex features from different CT dose labels. In the fine-tuning phase on our emulated data, the model learns complex CT image features with noise and also anatomical maps. Experimental evaluation on IQA and liver segmentation demonstrates the realistic quality variation and preservation of anatomical structures, along with noise-awareness in the generated CT images. While higher-dose and low-noise CT scans are preferable for diagnosis, noise variation itself is clinically important for downstream tasks such as CT IQA, dose optimization, and selecting scans suitable for radiologist review or downstream AI models. These tasks require both high-quality and low-quality examples to accurately identify when a scan is too noisy, low-dose, or degraded. Therefore, modeling noise variations is necessary to teach the generative model how CT appearance changes under different dose levels, which is essential for generating realistic low-quality images and for downstream dose- or quality-aware applications. Our noise-aware and anatomically consistent design enables NA-Diff to explicitly learn this dose-quality relationship with the focus on the target organ, rather than collapsing toward only high-quality reconstructions. Our future work will focus on a more extensive evaluation of NA-Diff across different CT organs and various downstream tasks.

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

## A  NA-DIFF VARIANTS

### A.1  SEP-DIFF

In Sep-Diff, it has three stages of training. In the first stage, the model uses a CT-emulated natural image and input conditioned on noise maps. This stage is considered a natural image pre-training phase. Both the input image and the output image are encoded using the VAE Kingma et al. (2019) encoder. In the second stage, the model is fine-tuned, leveraging CT images and their corresponding simulated noise maps. In this stage, the input image is also encoded using the VAE encoder; however, for noise maps, we use the CLIP Radford et al. (2021) model encoder to get a global semantic embedding as shown in Figure 5. In the final stage, the model fine-tunes utilizing CT images and their corresponding liver segmentation mask. For the image encoding process, the model follows a process similar to the second stage.

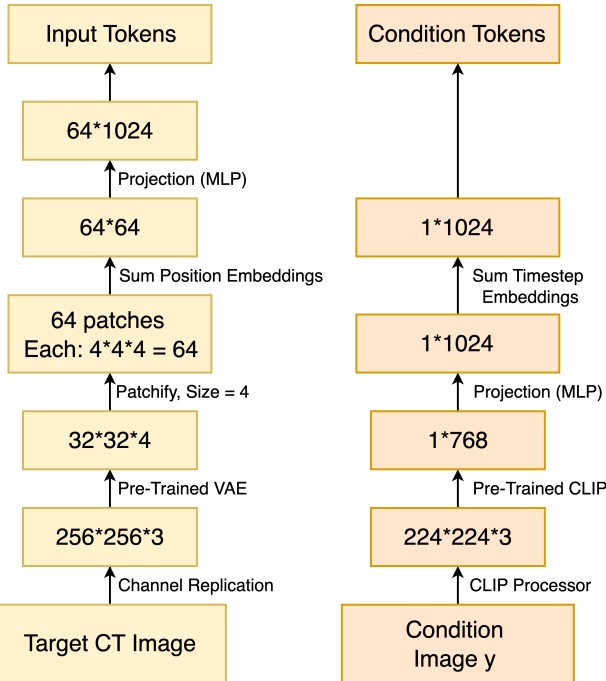

Figure 5: Dimensionality flow of Sep-Diff variants (excluding pre-training). The CT target image is encoded by a pre-trained VAE into latent patches (e.g., 4×4×4), flattened and projected to the transformer's hidden space. In parallel, the condition image is processed by a pre-trained CLIP model to produce a global feature vector, which is linearly projected, fused with timestep embeddings, and combined with input tokens for the transformer.

### A.2  CAT-DIFF

Unlike Sep-Diff, Cat-Diff contains a two-stage training process. In the first stage of training, it follows the exact same process as Sep-Diff pre-training. However, in the second stage, it combines the second and third stage training of Sep-Diff into one stage training. Here, the model leverages CT images with corresponding noise maps and anatomy masks as conditioning. While input images are encoded using VAE, both conditions use CLIP as an encoder. After getting the global embedding, we employ concatenation to emphasize dual conditioning. Let $y_{\text{map}}, y_{\text{mask}} \in \mathbb{R}^{768}$ denote the CLIP embeddings of the noise map and segmentation mask, respectively. These embeddings are projected into the transformer's hidden space using a learnable MLP:

$$\tilde{y}_{\text{map}} = \phi(y_{\text{map}}), \quad \tilde{y}_{\text{mask}} = \phi(y_{\text{mask}}), \quad \rho : \mathbb{R}^{768} \to \mathbb{R}^{\mathcal{D}}, \tag{10}$$

where $\mathcal{D}$ is the transformer hidden dimension (e.g., 1024), and $\rho$ is defined as:

$$\rho(y) = \mathcal{W}_2 \cdot \text{SiLU}(\mathcal{W}_1 y + \mathcal{B}_1) + \mathcal{B}_2. \tag{11}$$

To preserve the distinct information from both embeddings, we concatenate them along the channel dimension:

$$\tilde{y}_{\text{concat}} = [\tilde{y}_{\text{map}} \| \tilde{y}_{\text{mask}}] \in \mathbb{R}^{2\mathcal{D}}. \tag{12}$$

The concatenated vector is then fused and projected back to $\mathbb{R}^{\mathcal{D}}$ using a second MLP:

$$\tilde{y}_{\text{final}} = \eta(\tilde{y}_{\text{concat}}), \quad \eta : \mathbb{R}^{2\mathcal{D}} \to \mathbb{R}^{\mathcal{D}}. \tag{13}$$

Finally, this fused conditioning vector is added to the sinusoidal timestep embedding $\gamma(t)$ to form the complete conditioning representation:

$$c = \tilde{y}_{\text{final}} + f_\theta(t), \tag{14}$$

where $f_\theta(t) \in \mathbb{R}^{\mathcal{D}}$ is the output of an MLP applied to a sinusoidal timestep embedding. This conditioning vector $c$ is broadcast to the transformer blocks and modulates each residual path using adaptive layer normalization (adaLN-Zero).

## B  IQA AND SEGMENTATION EVALUATION

NA-Diff model variants are evaluated by predicting diagnostic image quality assessment (IQA) scores, providing a measure of noise level estimation for the generated CT images. Although the IQA score may not be perfectly accurate due to differences in window center (40) and width (400) settings between our dataset and the pre-trained IQA model's Rifa et al. (2025a) training data (as shown in Fig. 6), a relative comparison is still feasible. We utilize another pretrained model Ward & Imran (2024) for liver segmentation masks comparison; the generated CT images from our model are passed through the segmentation model to produce liver masks and show a comparison between the model-generated mask and the real mask images. This allows us to assess whether the anatomical structures are being effectively preserved in the synthesized images. We report Dice score, IoU, Precision, Recall, and Hausdorff Distance (HD) to assess how well the segmentation masks predicted from the generated CT images compare to the real ground-truth masks.

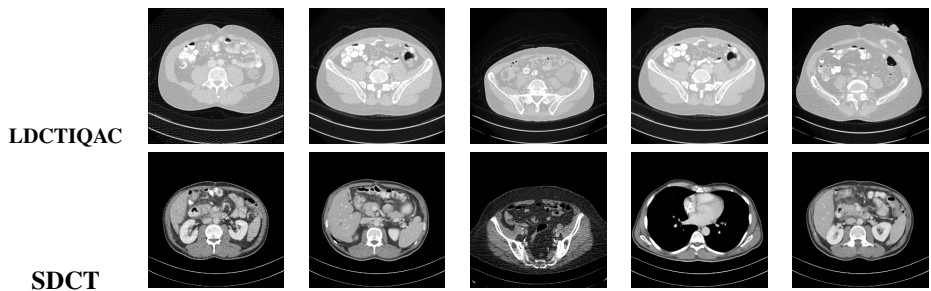

Figure 6: Visualization of the samples of IQA-based model's Rifa et al. (2025b) trained dataset LDCTIQAC Lee et al. (2023) and our SDCT dataset.

## C  ADDITIONAL EXPERIMENTS

The IQA scores for NA-Diff-D and NA-Diff-F are quite similar across different dose levels as shown in Fig. 7. For NA-Diff-F, the IQA scores increase steadily as the dose level rises, indicating that it is more effective at handling noise in generated images. Qualitative results comparing the baseline model and our anatomy-guided variants are shown in Fig. 8. Our method preserves anatomical structures more compared to the baseline.

Moreover, as reported in Table 8, our NA-Diff-D model outperforms the baseline when evaluated on the LiTS dataset. Compared to the Seg-Diff* baseline, our NA-Diff-D model achieves the best performance across all metrics, improving CFID by 11.98%, KID by 25%, CMMD by 11.4%, and LPIPS by 0.4%. This highlights the efficacy of our anatomy-guided approach for generating high-fidelity and perceptually consistent CT images. Also, the results in Table 9 highlight the advantage of concatenating condition tokens over summing them for dual conditioning.

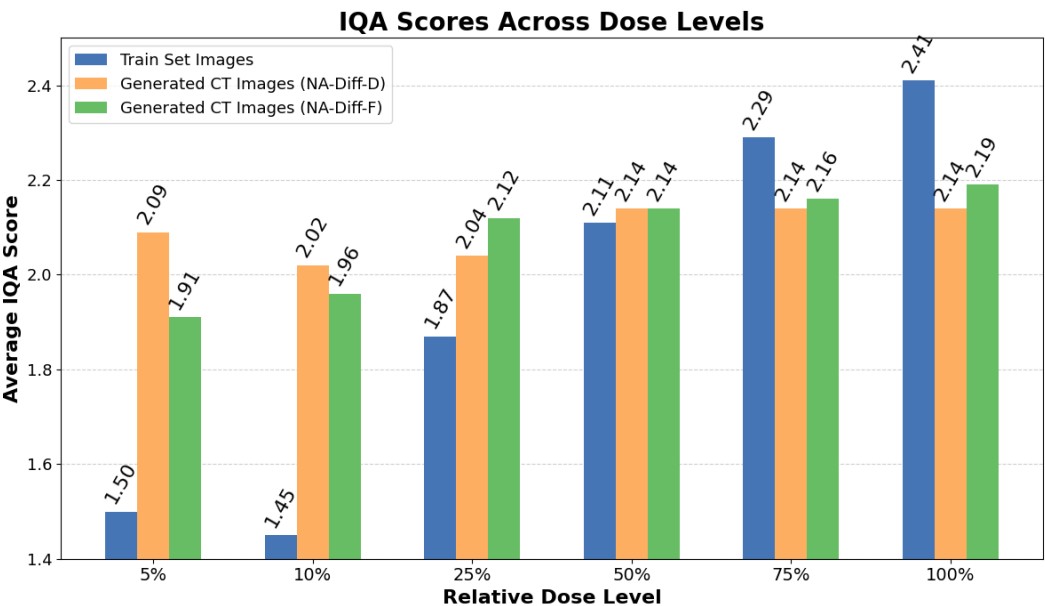

Figure 7: Predicted IQA scores for the NA-Diff-F generated CT images compared to input CT images at different dose levels.

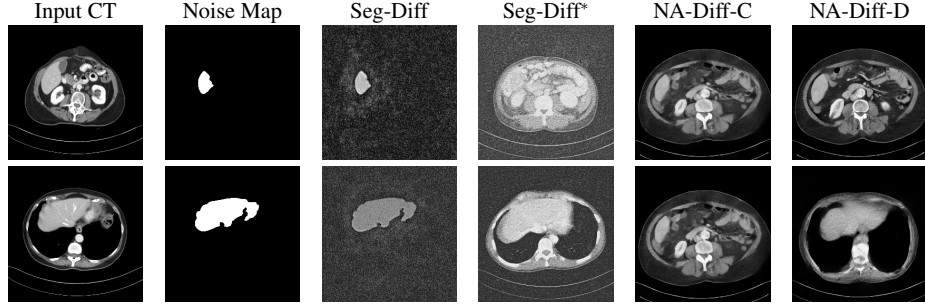

Figure 8: Qualitative results with baseline and antamody guided variants of NA-Diff.

In Fig. 10, NA-Diff-F demonstrates strong noise map awareness, adapting its generation quality following varied noise map levels. Furthermore, we have the overlay results from NA-Diff-D and NA-Diff-F as shown in Fig. 9, which show good anatomical alignment between the generated images and the segmentation masks.

Algorithm 2 shows how the input data is conditioned. The grayscale image $x$ is first repeated across three channels and encoded into a latent $z$ using the VAE encoder $\mathcal{E}$. The anatomy map $n$ and mask $m$ are separately encoded by the CLIP image encoder $\phi$ to obtain feature vectors $f_n$ and $f_m$. These are projected through $W_{\text{clip}}$ and fused by $W_{\text{cat}}$ to form the condition embedding $c$.

Algorithm 1 describes the DiT forward pass with adaLN-Zero conditioning. The latent $z$ is patch-embedded with fixed sinusoidal positional embedding $P$. A timestep $t$ is mapped into an embedding $e_t$ (via a sinusoidal MLP) and added to $c$ to form the global conditioner $h$. In each DiT block, $h$ is used by adaLN to generate scale $\Gamma$, shift $\Delta$, and gating $g$ parameters that modulate the LayerNorm outputs before multi-head self-attention (MSA) and MLP layers. After $L$ such blocks, the final layer produces the predicted noise $\hat{\epsilon}_\theta$. This prediction is used in the diffusion loss to train the model.

## D  CT NOISE SIMULATION

Diffusion models require large-scale training data. However, using natural images for pre-training can lead to domain shift issues. To address this, we emulate CT-specific noise patterns within natural images, creating CT dose-emulated datasets for the pre-training phase. Fig. 11 illustrates examples

Input CT     Anatomy Map     NA-Diff-D     NA-Diff-F

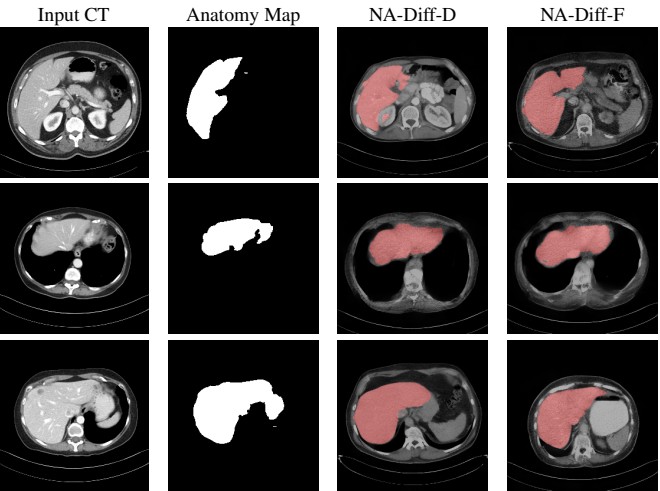

Figure 9: Qualitative comparison showing overlay outputs from our NA-Diff-D and NA-Diff-F variants. The visualization highlights segmentation accuracy and generative performance.

5% Dose     10% Dose     25% Dose     50% Dose     75% Dose     100% Dose

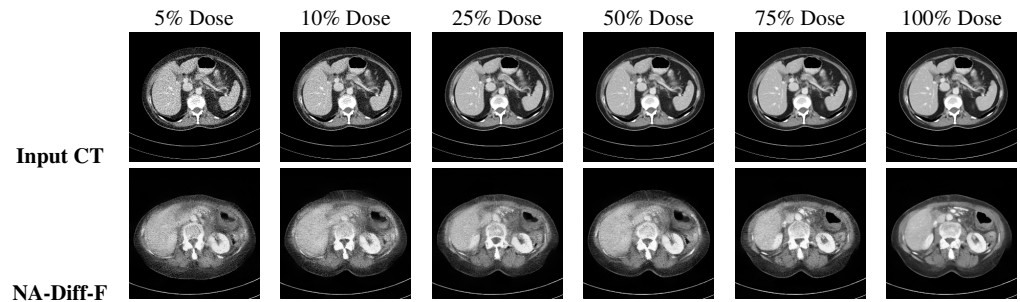

Figure 10: Qualitative comparisons at different noise levels show that our NA-Diff-F model effectively captures and adapts to the provided noise maps.

Table 8: The models are fine-tuned (with 3000 random sample images) and evaluated (with 500 random sample images) with segmentation guidance using the LiTS Bilic et al. (2023) dataset. Our synthetic datasets show superior performance to the baseline. ∗ denotes that the model is trained with our CT image data conditioned on the anatomy mask. The best and second-best results are **bolded** and underlined respectively.

| Methods | Performance metrics | | | |
|---|---|---|---|---|
| | CFID ↓ | KID ↓ | CMMD ↓ | LPIPS ↓ |
| Seg-Diff* | 60.5636 | 25.3853 ± 3.065 | 0.376 | 0.3833 ± 0.1023 |
| NA-Diff-C | 89.2949 | 57.6014 ± 5.8231 | 0.541 | 0.3866 ± 0.1021 |
| NA-Diff-D | **53.2949** | **19.0386 ± 2.8532** | **0.333** | **0.3817 ± 0.1029** |

Table 9: Ablation study highlighting the importance of using concatenation in dual conditioning. The sum and concatenation denotes operations on condition token.

| Methods | Operation | Performance metrics | | | |
|---|---|---|---|---|---|
| | | CFID ↓ | KID ↓ | CMMD ↓ | LPIPS ↓ |
| NA-Diff-F | Sum | 81.3672 | 52.7426 ± 5.2481 | 0.492 | 0.3824 ± 0.1018 |
| NA-Diff-F | Concat. | **64.6379** | **24.8106 ± 3.2275** | **0.3205** | **0.3760 ± 0.0994** |

of natural images augmented with CT-like noise characteristics. Moreover, Fig. 12 visualizes simulated CT images corresponding to relative dose levels.

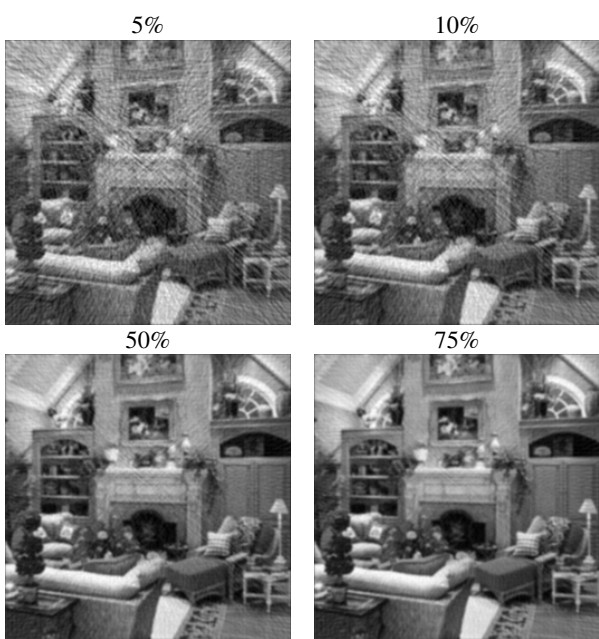

Figure 11: Visualization of natural images at different simulated dose levels.

---

**Algorithm 1** DiT forward pass with AdaLN-Zero

---

**Require:** latent image tokens $z$, condition embedding $c$, timestep embedding $t$

1: $X \leftarrow \texttt{PatchEmbed}(z) + P$                ▷ initial token sequence

2: $h \leftarrow \text{sincosMLP}(t) + c$               ▷ conditioning vector

3: **for** $\ell = 1$ to $L$ **do**

4:     $(\gamma_t^{(\ell)}, \beta_t^{(\ell)}, \alpha_t^{(\ell)}) \leftarrow \texttt{MLP}(\texttt{LN}(h))$

5:     $X \leftarrow X + \alpha_t^{(\ell)} \cdot \texttt{MSA}\left(\gamma_t^{(\ell)} \odot \texttt{LN}(X) + \beta_t^{(\ell)}\right)$

6:     $X \leftarrow X + \alpha_t^{(\ell)} \cdot \texttt{MLP}\left(\gamma_t^{(\ell)} \odot \texttt{LN}(X) + \beta_t^{(\ell)}\right)$

7: **end for**

8: $(\gamma_t', \beta_t') \leftarrow \texttt{MLP}(\texttt{LN}(h))$

9: $\hat{\vartheta} \leftarrow \texttt{unpatchify}\left(W_{\text{out}}\left(\gamma_t' \odot \texttt{LN}(X) + \beta_t'\right)\right)$

10: **return** $\hat{\vartheta}$               ▷ predicted noise

---

**Algorithm 2** Data conditioning

---

**Require:** $x$ (gray), $n$ (map), $m$ (mask); $\mathcal{E}, \phi$

1: $z \leftarrow 0.18215 \cdot \mathcal{E}(\text{repeat}(x, 3)).\texttt{sample()}$

2: $f_n \leftarrow \phi(n); \ f_m \leftarrow \phi(m)$

3: $c \leftarrow W_{\text{cat}}\left(\left[W_{\text{clip}}(f_n) \parallel W_{\text{clip}}(f_m)\right]\right)$

4: **return** $(z, c)$

---

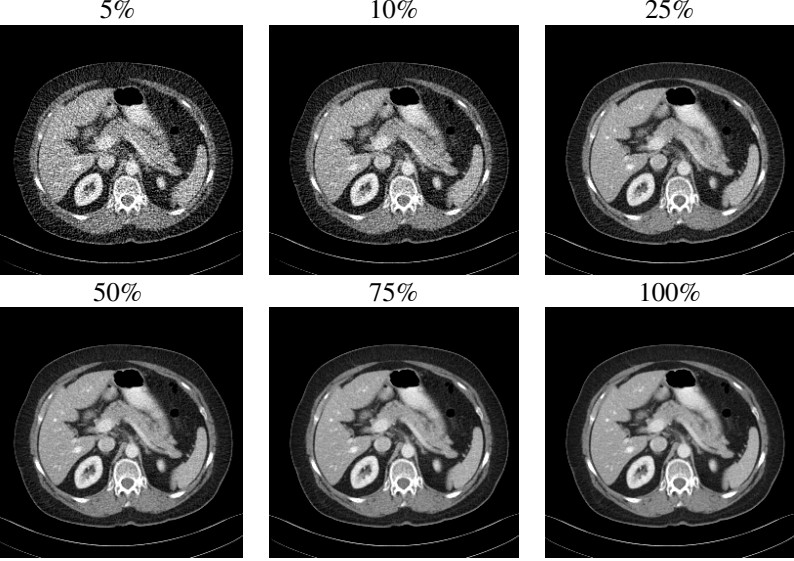

Figure 12: Visualization of CT images across different simulated dose levels (e.g., only 25% and 100% are provided; other levels are constructed using our low-dose simulation technique).

