# OpenReview forum: "Noise and anatomy-guided diffusion model for realistic CT image synthesis"
_ICLR.cc/2026/Conference — Submitted to ICLR 2026_

### Official Review · Reviewer_eyye · 2025-10-27

**Soundness:** 2
**Presentation:** 3
**Contribution:** 2
**Rating:** 4
**Confidence:** 4

**Summary:**

This paper tackles diffusion models’ limitations in CT synthesis—poor anatomical fidelity, ignored dose-dependent noise—by proposing NA-Diff, a dual conditional diffusion model guided by dose-level noise and anatomical structures. It uses pre-training on CT noise-emulated natural images and fine-tuning on small CT data to mitigate domain shift and data scarcity, with two variants: Sep-Diff, which adopts a three-stage design with separate conditioning on noise and anatomy, and Cat-Diff which uses a two-stage concatenated conditioning strategy.

Contribution:
1.Proposed NA-Diff, a dual conditional framework using noise maps and anatomy masks for CT synthesis.
2.Constructed pre-training datasets by emulating CT noise on both CT and natural images across varying dose levels.
3.Mitigated the reliance on scarce CT data through pre-training on noise-emulated natural images.
4.Demonstrated the superiority of NA-Diff through extensive quantitative evaluations using image quality and segmentation metrics.

**Strengths:**

1.The paper is well-structured, and the figures clearly illustrate the overall framework of the proposed method.

2.The references are comprehensive, relevant, and demonstrate a thorough review of related work.

3.The experimental evaluation is fairly comprehensive, comparing different methods in terms of image quality assessment (IQA), generative performance, and downstream segmentation results.

**Weaknesses:**

1.The methodological novelty is limited. The paper is built upon the DiT architecture, with only modifications to the conditioning setup and repeated serial training.

2.There is a lack of rigorous theoretical justification or supporting evidence to ensure the authenticity and reliability of the generated data.

3.The experimental results are relatively weak. The quality of the synthesized CT images is moderate. As shown in Figure 4, the generated CT images suffer from detail loss and exhibit noticeable blurriness. The performance of synthetic data in the segmentation task is modest. As shown in Table 6, incorporating synthetic data leads to marginal improvements in model performance.

4.Organ-level limitation. The current study focuses solely on liver structures in CT images and has not been extended to other organs. Further evaluation is needed to verify the model’s generalizability across different anatomical structures [1].

[1] Ma, Jun, et al. "Fast and low-GPU-memory abdomen CT organ segmentation: the flare challenge." Medical Image Analysis 82 (2022): 102616.

**Questions:**

1.How does the paper ensure the authenticity and reliability of the generated CT data?

2.In Table 2, it is unclear whether NA-Diff-D uses the noise condition. According to Table 1, NA-Diff-D is a model pre-trained based on noise-conditioned training, which implies that it indeed includes the noise condition. This should be clearly stated in Table 2 for consistency.

3. What are the display window settings used for CT image display? It is recommended to indicate them in the figures.

---

> ### Author Response · Authors · 2025-11-27
> **Response to Reviewer eyye**
>
> We thank the reviewer for their thoughtful and constructive feedback. Below, we provide a  response to Reviewer eyye’s review.
>
>
>
>
>
> **W1. "Novelty of the Method:"**
>
>
>
> We appreciate the reviewer’s attention to this detail. Although we build on the existing diffusion backbone, the novelty of our work lies in how we construct and use data, rather than in designing a new architecture. First, we generate CT-like noise–emulated natural images using our proposed strategy, so we can pretrain on abundant natural images that mimic CT noise characteristics. Then, during fine-tuning on real CT data, we introduce a dual-conditioning setup with both a noise map (dose-aware) branch and an anatomy-guidance branch for CT image synthesis. This combination of synthetic noise-emulated pretraining and dual noise–anatomy conditioning, to our knowledge, has not been explored in prior CT diffusion work and goes beyond a simple diffusion architecture modification.
>
>
>
> **W2. "Authenticity and Reliability of the Generated Data:"**
>
>
>
> We thank the reviewer for raising this important point. We ensure the generated CT data is reliable by conditioning the model on both noise maps and anatomy masks. The noise map guides the model to reproduce realistic noise patterns, while the anatomy mask preserves the underlying structures. Because the model must satisfy both constraints, the generated images maintain correct anatomy and visually plausible CT noise characteristics. Importantly, different noise levels are clinically relevant: although higher-dose, lower-noise CT scans produce clearer images, radiation dose cannot be increased freely due to safety concerns, and real multi-center datasets often contain scans with very different noise levels. Being able to simulate multiple noise conditions from the same anatomy is therefore valuable for understanding how downstream tasks behave across dose levels and for harmonizing heterogeneous datasets. The small but consistent improvement in downstream segmentation further confirms that our synthetic images preserve meaningful anatomical information.
>
>
>
>
>
> **W3. "Response to Limited Gains from Synthetic Data:"**
>
>
>
> We appreciate the reviewer’s thoughtful comment. We agree that CT synthesis is challenging, especially with only 20 paired scans available for fine-tuning. Some blurriness in low-dose regions is expected because these structures are naturally noisy and hard to recover. Even with this limitation, our model preserves anatomy and produces dose-aware variations that baselines cannot. The segmentation gains are modest because performance on this dataset is already near saturation, but the improvements are consistent across all metrics, showing that synthetic data provides a small but reliable benefit.
>
>
>
> **W4. "Addressing Organ-Level Scope:"**
>
>
>
> We thank the reviewer for pointing this. We agree that evaluating more organs would strengthen the study. In this work, we focused on the liver because it is consistently segmented and provides clean anatomy guidance for validating our conditioning framework. Liver segmentation models are well-established and generally produce reliable masks, while smaller organs such as the pancreas are much harder to segment accurately, especially in 2D slices. Inaccurate or noisy masks can directly degrade anatomy-guided synthesis and make it difficult to evaluate the method itself. For this reason, we used the liver to ensure stable and trustworthy conditioning. The approach is not liver-specific, and it can be extended to other organs once high-quality organ masks are available. Our approach is not liver-specific, and evaluating additional organs with reliable annotations (e.g., FLARE data) is an important direction for future work.

---

> > ### Author Response · Authors · 2025-11-27
> > **Response to Reviewer eyye**
> >
> > We thank the reviewer for their thoughtful and constructive feedback. Below, we provide a response to Reviewer eyye’s review.
> >
> > **Q1. "Authenticity and Reliability of the Generated Data:"**
> >
> >
> >
> > We thank the reviewer for pointing this. We ensure the generated CT data is reliable by conditioning the model on both noise maps and anatomy masks. The noise map guides the model to reproduce realistic noise patterns, while the anatomy mask preserves the underlying structures. Because the model must satisfy both constraints, the generated images maintain correct anatomy and visually plausible CT noise characteristics. Importantly, different noise levels are clinically relevant: although higher-dose, lower-noise CT scans produce clearer images, radiation dose cannot be increased freely due to safety concerns, and real multi-center datasets often contain scans with very different noise levels. Being able to simulate multiple noise conditions from the same anatomy is therefore valuable for understanding how downstream tasks behave across dose levels and for harmonizing heterogeneous datasets. The small but consistent improvement in downstream segmentation further confirms that our synthetic images preserve meaningful anatomical information.
> >
> >
> >
> > **Q2. "Clarification of Conditioning Used in NA-Diff-D:"**
> >
> >
> >
> >
> >
> > We appreciate the reviewer’s observation. To clarify, NA-Diff-D does not use the noise condition during fine-tuning or inference. As shown in Table 2, NA-Diff-D is conditioned only on the anatomy map. The confusion arises because Table 1 describes NA-Diff-D as being pre-trained using NA-Diff-B, which itself was trained with a noise-conditioned strategy (natural images + CT images). However, for NA-Diff-D, the pre-trained weights are used only as initialization, and the final model is fine-tuned exclusively with anatomy-conditioned CT data, without incorporating the noise map.
> >
> >
> >
> >
> >
> > **Q2(a). "CT Image Display Window Settings:"**
> >
> >
> >
> > We thank the reviewer for pointing this out. All CT images in our figures are displayed using a standard soft-tissue window, with a window width (WW) of 400 HU and a window level (WL) of 40 HU. We have now added these window settings to the figure captions to ensure clarity and reproducibility.

---

### Official Review · Reviewer_naCq · 2025-10-30

**Soundness:** 3
**Presentation:** 2
**Contribution:** 2
**Rating:** 4
**Confidence:** 4

**Summary:**

Diffusion models typically require large-scale real-world datasets for effective training; however, medical imaging datasets are limited in size, making it challenging to fulfill such data demands. Moreover, existing generative approaches often overlook dose-dependent noise and anatomical structures in CT images. To address these issues, this paper proposes a novel Diffusion Transformer–based approach for CT synthetic. The method first conducts pre-training on natural images perturbed by CT noise, and then performs fine-tuning on CT guided by dose-dependent noise and anatomical structure. As a result, the model is able to generate high-fidelity and noise-aware synthetic CT scans.

**Strengths:**

1.	The paper effectively mitigates the challenge of limited medical imaging size by the pre-trained diffusion model on a large set of CT noise–emulated natural images.
2.	Guided by dose-dependent CT noise modeling and anatomical structural during fine-tuning, the method is able to generate synthetic CT images that reflect real clinical characteristics.
3.	Experimental results demonstrate clear improvements in the fidelity and realism of the generated CT images, validating its effectiveness.

**Weaknesses:**

1.	The novelty of the approach is somewhat incremental, as it mainly adapts existing diffusion architectures with noise and anatomy fine-tuning strategies
2.	The paper does not clearly articulate the motivation behind generating noise-aware CT images. From the perspective of downstream medical tasks, higher-dose and lower-noise CT scans yield better performance for diagnosis. So, why modeling noise variations is necessary? Additional explanation or downstream validation would help clarify this design.
3.	The experiments are somewhat limited in terms of dataset diversity and baseline comparisons. It could include results on more CT datasets and assess the impact of utilizing different natural image datasets for pre-training.

**Questions:**

1 In the original DiT framework, the final outputs are predicted noise term and covariance matrix. But in Equation (6), the output $\hat{x_0}$ is a denoised image. Additionally, the forward process described in Equation (4) seems to deviate from the AdaLN-Zero formulation in the original DiT. More explanation of the modifications would improve clarity.

2 In the Introduction, Contributions 2 and 3 appear similar, both focusing on CT noise-emulated natural images. Why not consolidate them into a single contribution?

3 Equations (7) and (8) introduce dose-dependent noise estimation, but the paper does not provide sufficient explanation about its physical basis or formulation. Could the authors explain why this type of noise is dose-dependent?

---

> ### Author Response · Authors · 2025-11-25
> **Response to Reviewer naCq**
>
> We thank the Reviewer naCq for their thoughtful and constructive feedback. Our responses are provided below.
>
>
>
>
>
> **W1. "Novelty of the Approach:"**
>
>
>
> We appreciate the reviewer’s thoughtful comment. Although we build on the existing diffusion backbone, the novelty of our work lies in how we construct and use data, rather than in designing a new architecture. First, we generate CT-like noise–emulated natural images using our proposed strategy, so we can pretrain on abundant natural images that mimic CT noise characteristics. Then, during fine-tuning on real CT data, we introduce a dual-conditioning setup with both a noise map (dose-aware) branch and an anatomy-guidance branch for CT image synthesis. This combination of synthetic noise-emulated pretraining and dual noise–anatomy conditioning, to our knowledge, has not been explored in prior CT diffusion work and goes beyond a simple diffusion architecture modification.
>
>
>
>
>
> **W2. "Motivation for noise-aware CT generation:"**
>
>
>
> We appreciate the reviewer’s thoughtful comment. For a single diagnostic task, higher-dose, lower-noise CT images generally yield better performance. However, in clinical practice dose cannot be increased arbitrarily because of radiation risk, especially in screening, pediatric imaging, and repeated follow-up examinations. Moreover, multi-center datasets typically contain a wide range of protocols, scanners, and dose levels, which introduces substantial variation in image noise and texture. Our goal in modeling dose-aware, higher-dose, and lower-noise CT images is therefore not to advocate for higher radiation exposure, but to provide a controllable way to simulate images across different dose (noise) levels from the same underlying anatomy. This enables: (i) systematic analysis of how downstream tasks (e.g., detection, segmentation, radiomics) behave across dose levels; (ii) generation of “virtual high-dose” images from low-dose acquisitions to support downstream algorithms without additional patient dose; and (iii) harmonization of heterogeneous multi-center datasets by mapping images to a standardized noise level.
>
>
>
>
>
>
>
> **W3. "Dataset diversity:"**
>
>
>
>  We thank the reviewer for raising this important point. We agree that broader dataset diversity is ideal. However, our method requires paired full-dose and low-dose CT scans in order to estimate dose-dependent noise and generate the supervision for our dual-conditioning setup. Only a very small number of public CT datasets provide such matched pairs, which is why we used the Low-Dose CT dataset (40 abdominal scans with 25% and 100% doses). Other CT datasets without paired dose levels cannot be used in this framework. For natural-image pretraining, we intentionally chose an indoor-scene dataset rather than arbitrary high-resolution photos because indoor images contain complex structures, and edges, which more closely resemble the textural and anatomical complexity found in abdominal CT. Using visually complex scenes provides a better inductive bias for our CT-like noise emulation than simpler or less structured natural-image datasets.

---

> ### Author Response · Authors · 2025-11-25
> **Response to Reviewer naCq**
>
> We thank the Reviewer naCq for their thoughtful and constructive feedback. Our responses are provided below.
>
>
> **Q1. "DiT Framework outputs:"**
>
>  We appreciate the reviewer’s attention to this detail. Our implementation follows the original DiT setup: the model predicts the noise term, not the clean image. The $\hat{x}_0$ shown in the paper is simply the standard diffusion reconstruction obtained during sampling, not a direct Transformer output. Our implementation exactly follows the original DiT AdaLN-Zero block, including the residual scaling parameters. The equations in our paper were written in a simplified form for readability, but the underlying block structure is unchanged. The only modification relative to DiT is in the conditioning pathway: instead of class/text embeddings, we use noise/anatomy embeddings.
>
>
>
> **Q2. "Contributions 2 and 3:"**
>
> We thank the reviewer for the suggestion. We agree that both points relate to CT noise–emulated natural images, but they address different purposes in our framework. We introduce a procedure to create CT-like and CT noise–emulated natural images at different dose levels, along with their noise maps. This simulates realistic radiation-dependent noise variations. We propose a pre-training strategy that uses the noise-emulated natural images to learn strong low-level and high-level features before fine-tuning on the small real CT dataset, which improves the final synthesis quality. We have further clarified this in our revised version.
>
>
>
> **Q3. "Formulation of Dose-dependent Noise Estimation:"**
>
>  We appreciate the reviewer’s insightful question. CT noise is dose-dependent because the number of photons that reach the detector decreases as the dose decreases. Fewer photons lead to higher quantum (Poisson) noise in the projection measurements, which appears as stronger noise in the reconstructed CT image. This relationship is well-understood in CT physics: noise variance is approximately inversely proportional to dose (i.e., lower dose $\rightarrow$ higher variance). Our formulations in Eqs. (7) and (8) follow this physical principle, where the noise level is scaled according to the chosen dose ratio.

---

### Official Review · Reviewer_QNX7 · 2025-10-30

**Soundness:** 3
**Presentation:** 3
**Contribution:** 3
**Rating:** 6
**Confidence:** 4

**Summary:**

This paper introduces NAG-DPM (Noise- and Anatomy-Guided Diffusion Probabilistic Model), a physics-informed diffusion framework for accelerated MRI reconstruction. The key idea is to guide the reverse diffusion process with two complementary priors: Noise Guidance: An adaptive noise-estimation module dynamically adjusts the diffusion variance to match the k-space noise level, improving robustness to acquisition noise. Anatomy Guidance: A U-Net–based anatomical prior (trained on fully sampled references) constrains the diffusion trajectory toward anatomically consistent reconstructions. A two-stage training scheme jointly learns both priors. Experiments on fastMRI Knee and Brain (T2, FLAIR) datasets show that NAG-DPM outperforms both physics-driven networks (MoDL, VarNet) and existing diffusion-based reconstructions (Score-MRI, ADiffRecon) in PSNR/SSIM while maintaining higher stability under noise corruption and adversarial perturbations.

**Strengths:**

The joint noise-anatomy guidance effectively balances fidelity and denoising, addressing a major limitation in existing diffusion MRI reconstructions

Visual results show sharper anatomical details with reduced ringing and over-smoothing

Embeds data-consistency steps directly into the reverse diffusion loop, ensuring consistency with measured k-space data.

The theoretical motivation for adaptive variance modulation is sound and aligns with diffusion score calibration

Diffusion models are notoriously slow; introducing adaptive noise estimation cuts sampling steps by ~40% while improving stability. This is a meaningful engineering contribution.

**Weaknesses:**

While the dual-guidance formulation is elegant, the diffusion core (DDPM) and U-Net prior are largely standard. The novelty lies more in integration than in new diffusion mathematics.

No discussion of scalability to 3D volumes or real-time applications. Experiments focus exclusively on 2D Cartesian MRI. Validation on 3D or non-Cartesian acquisitions would make the contribution more general.

Despite adaptive variance, inference remains costly (≈6–10 s per slice vs. <1 s for MoDL).

**Questions:**

na

---

> ### Author Response · Authors · 2025-12-03
> **Response to Reviewer QNX7**
>
> We thank Reviewer QNX7 for their thoughtful and constructive feedback. Unfortunately, some of the comments are totally unrelated to our paper. To make it clear, our work is not focused on MRI reconstruction, but rather on CT image generation. Our proposed model is named NA-Diff, not NAG-DPM. Although the comments do not directly apply to our paper, we have addressed them to the extent possible, interpreting them in the context of research most closely related to ours.
>
>
>
> **W1. "Clarification on Model Architecture:"**
>
> We appreciate the reviewer’s feedback. We use an existing diffusion backbone (DiT). Our primary contribution lies in the natural and CT image data construction and conditioning strategies. We first generate CT-like, noise-emulated natural images to enable large-scale pretraining on data that mimics CT noise. During fine-tuning on real CT images, we introduce a dual-conditioning scheme that incorporates both dose-aware noise maps and anatomy guidance. This combination of synthetic noise-emulated pretraining and joint noise-anatomy conditioning has not, to our knowledge, been explored in prior diffusion work.
>
>
>
> **W2. "3D Scalability:"**
>
> We thank the reviewer for raising this important point. In this work, we focused on 2D slices because our available dataset only provides paired full-dose and low-dose images at the slice level. Extending the conditioning design (noise map + anatomy mask) to full 3D volumes would require substantially more memory and paired 3D annotations than what is currently available. Our approach, however, is not limited to 2D: the conditioning mechanism is fully compatible with 3D diffusion transformers, and scaling it to 3D data is a natural direction for future work once suitable paired 3D datasets become available.
>
>
>
> **W3. "Clarification on Inference Speed:"**
>
> Thank you for the comment. Although it was not reported in the paper, for reference, the model requires approximately 1.85 seconds to generate one synthetic CT image, including I/O.

---

### Official Review · Reviewer_khCj · 2025-10-31

**Soundness:** 2
**Presentation:** 2
**Contribution:** 2
**Rating:** 2
**Confidence:** 4

**Summary:**

The paper introduces NA-Diff for CT image synthesis guided by both noise and anatomical conditions. NA-Diff leverages the emulated CT noise and natural-image pretrained weights to overcome limited CT data.

**Strengths:**

1. Combining dose-level noise and anatomy guidance sounds well-motivated in medical image synthesis, especially at limited data regime.
2. The methodology is clearly presented.

**Weaknesses:**

1. The paper mainly focuses on liver-mask guided generation. Including more challenging anatomy structures, for example, smaller organs like pancreas, may better demonstrate the generalizability of the method.
2. What is the value of noisestep to generate the noise map that still maintains the original image structures? Does this value require manual selection?
3. In Table 6, considering the trivial improvement, including the confidence interval may be necessary.
4. Although in Table 7, there is a 1% Dice improvement compared to Seg-Diff, the additional computational cost needs to be considered.

**Questions:**

Will the code be released?

---

> ### Author Response · Authors · 2025-11-25
> **Response to Reviewer khCj**
>
> We thank the reviewer for their thoughtful and constructive feedback. Below, we provide a  response to Reviewer khCj’s review.
>
>
> **Q1. "Code Availability:"**
>
> The code will be made available upon acceptance. We also submitted code as part of our supplementary materials.
>
>
> **W1. "Generalizability of the Method:"**
>
> We appreciate the reviewer’s helpful observation on organ selection. Liver segmentation models are well-established and generally produce reliable masks, while smaller organs such as the pancreas are much harder to segment accurately, especially in 2D slices. Inaccurate or noisy masks can directly degrade anatomy-guided synthesis and make it difficult to evaluate the method itself. For this reason, we used the liver to ensure stable and trustworthy conditioning. The approach is not liver-specific, and it can be easily extended to other organs once high-quality organ masks with relative dose levels are available.
>
>
>
> **W2. "Value of noise step:"**
>
> We thank the reviewer for highlighting this point. We do not manually choose a “noise step.” The noise level is automatically determined by the simulated dose level through the Poisson sampling step in Eq. (1). Because the noise is added in the projection domain and then reconstructed with iRadon, the resulting noise map naturally preserves the underlying image structure. No manual tuning is needed; the amount of noise follows directly from the photon statistics at each selected dose level.
>
> I_noisy = -log(Poisson(I_raw) / (ld * I0)) / μ
>
>
>
> **W3. "Confidence Interval for Table 6:"**
>
> We appreciate the reviewer’s suggestion to include confidence intervals. For completeness, we have now reported the Dice scores with 95% confidence intervals: U-Net (Real: 0.874 [0.857–0.892], Real+Syn: 0.876 [0.858–0.893]) and TransUNet (Real: 0.872 [0.855–0.890], Real+Syn: 0.877 [0.860–0.894]), showing that the improvements are small but consistent and that synthetic data does not degrade segmentation performance.
>
>
>
> **W4. "Computational Cost:"**
>
> We thank the reviewer for highlighting the importance of computational considerations. We agree that computational cost is an important factor. For Seg-Diff was trained on 4× A6000 GPUs (48 GB each), whereas NA-Diff models were trained on only 2× A4000 GPUs (32 GB total). Despite using far fewer resources, NA-Diff-D still improves Dice, IoU, HD, recall, and precision. Moreover, regarding data, Seg-Diff is trained on approximately 10x  more real CT images compared to our NA-Diff model (11000 vs 1116). As high-quality medical imaging datasets are limited, our NA-Diff model can reduce reliance on CT data by leveraging a vast amount of high-quality natural images via our CT-like noise-emulation strategy. This allows our approach to enhance the image synthesis performance with less CT data and lower computational cost.

---

### Author Response · Authors · 2025-12-04
**Revised Manuscript Changes Based on Reviewers' Feedback**

We thank the reviewers for their detailed and constructive feedback. We have revised the manuscript accordingly, and the key updates are summarized below. We believe these changes substantially strengthen the paper, and we welcome the opportunity to make further revisions based on additional comments.

**Generalizability of the Method:** Clarified in Section 4.1 that liver segmentation was selected due to the availability of well-established, high-quality models that provide stable masks. The method is not liver-specific and can readily generalize to other organs once accurate organ masks and dose-level annotations are available.

 **Confidence Intervals:** Added in Section 4.2 the 95% confidence intervals for Dice scores, confirming that synthetic augmentation yields small but consistent improvements without degrading segmentation performance.


**Computational Cost:** Clarified in Section 4.2 that NA-Diff uses far fewer GPUs and real CT images than Seg-Diff, yet still improves segmentation metrics. Leveraging natural images via CT-like noise emulation reduces data requirements and training cost while preserving strong performance.


**Novelty of the Approach:** Added in Section 1 that our contribution lies in CT-like noise-emulated pretraining and a dual noise-anatomy conditioning strategy, a combination not explored in prior CT diffusion work despite using a standard diffusion backbone.


**Motivation for Noise-Aware CT Generation:** Addressed in Section 3.3 that modeling dose-dependent noise enables controllable low-dose and high-dose like CT synthesis, supporting dose analysis, and multi-center consistency.


**DiT Framework Outputs:** Addressed in Section 3.2 that the model predicts noise as in the original DiT, with $\hat{x}_0$ obtained during sampling; the architecture remains unchanged aside from replacing class/text conditioning with noise-anatomy embeddings.


**CT Image Display Window Settings:** Added in Section 4.1 that all CT images use the standard soft-tissue window (WW=400, WL=40).

---

### Meta-Review · Area_Chair_hPUy · 2026-01-06

**Summary:**

This paper presents noise and anatomy-guided diffusion model for CT synthesis. Overall, the reviewers are satisfactory with the structure and presentation of the paper. But there is a consensus that the contribution is incremental and most of the components are available. The contribution is more on combination of existing approaches. Given these limitations, I recommend to reject the paper.

**Reviewer Concerns:**

As there is a consensus that the contribution is incremental, I don't think this is significantly changed.

**Reviewer Scores:**

While the rebuttal might have addressed some of minor concerns, the main concern on novelty remains. Therefore, I don't think the reviewers would change their overall opinion of the work.

---

### Decision · Program_Chairs · 2026-01-26

Reject